# Peripheral membrane protein endophilin B1 probes, perturbs and permeabilizes lipid bilayers

Arni Thorlacius ⓘ , Maksim Rulev, Oscar Sundberg ⓘ & Anna Sundborger-Lunna ⓘ ✉

Bin/Amphiphysin/Rvs167 (BAR) domain containing proteins are peripheral membrane proteins that regulate intracellular membrane curvature. BAR protein endophilin B1 plays a key role in multiple cellular processes critical for oncogenesis, including autophagy and apoptosis. Amphipathic regions in endophilin B1 drive membrane association and tubulation through membrane scaffolding. Our understanding of exactly how BAR proteins like endophilin B1 promote highly diverse intracellular membrane remodeling events in the cell is severely limited due to lack of high-resolution structural information. Here we present the highest resolution cryo-EM structure of a BAR protein to date and the first structures of a BAR protein bound to a lipid bicelle. Using neural networks, we can effectively sort particle species of different stoichiometries, revealing the tremendous flexibility of post-membrane binding, pre-polymer BAR dimer organization and membrane deformation. We also show that endophilin B1 efficiently permeabilizes negatively charged liposomes that contain mitochondria-specific lipid cardiolipin and propose a new model for Bax-mediated cell death.

Regulation of membrane remodeling is essential for maintaining cellular homeostasis[1]. Proteins from the Bin/Amphiphysin/Rvs167-homology (BAR) domain superfamily are peripheral membrane proteins that promote membrane curvature in eukaryotic cells[2–6]. BAR family members are grouped into classical BAR proteins, N-terminal amphipathic helix-BAR (N-BAR) proteins, BAR-pleckstrin homology (BAR-PH) proteins, Phox homology-BAR (PX-BAR) proteins, Fes/CIP4 homology-BAR (F-BAR) proteins, and inverse-BAR (I-BAR) proteins, based on domain organization and function[3,4]. BAR proteins are typically dimers in solution, where the dimer consists of three α-helices from each monomer that form a 6-helix bundle with a characteristic crescent shape. The concave surface of BAR domains is rich in amino acids with basic side chains and therefore, preferentially binds anionic membranes via electrostatic interactions[6–8]. BAR proteins are predominantly associated with helical scaffold assembly and the formation of membrane tubules[8–26].

Endophilins are a highly conserved group of N-BAR proteins that contain an N-terminal amphipathic helix (H0) and a 20-residue amphipathic insert in helix 1 (H1i)[8,10]. These two motifs contribute to membrane association[8,10]. A C-terminal Src-homology 3 (SH3) domain, which is connected to the BAR domain by a long flexible linker, mediates protein–protein interactions[8]. Endophilin B1 preferentially binds membranes that contain cardiolipin[25,27], a mitochondria-specific lipid that plays a critical role in Bax-mediated apoptosis[28]. Knockdown of endophilin B1 results in aberrant mitochondrial morphology and delayed Bax-mediated apoptosis[29,30]. Evidence suggests that endophilin B1 interacts with Bax via H0[27,31,32]. Interestingly, the H0 of endophilin B1 is longer than that of other endophilin family members (Table 2) and has a zero net charge, unlike the endophilin A1 H0, which is positively charged[25,33]. Endophilin B1 has also been shown to interact with Beclin-1 through UVRAG to promote autophagosome formation[34,35]. During mitophagy, it may interact with mitochondrial inner membrane protein prohibitin and form heterodimers with endophilin B2[36,37]. Loss of endophilin B1 is seen in several different forms of cancer, which indicates it plays an important tumor suppressor role in the cell[38–45].

The exact molecular mechanisms underlying the activity of endophilin B1 at intracellular membranes are unclear. Our previous cryo-electron microscopy (cryo-EM) studies of endophilin B1 reveal it organizes into helical scaffolds on tubulated liposomes that vary greatly in outer diameter (40–60 Å)[25]. This heterogeneity led to poor resolution at the protein-membrane interface and thus, poor insight into the organization of amphipathic regions H0 and H1i[25]. We further found that the flexible linker-SH3 domain region interacts with H0 in solution and that truncation of this region yielded more efficient liposome tubulation[25]. These findings suggest that endophilin B1 is autoinhibited in solution by intramolecular interactions between H0 and the SH3 domain. Similar BAR protein auto-inhibition by an SH3 domain was previously proposed for F-BAR protein syndapin-1[20].

Department of Cell and Molecular Biology, Uppsala University, Uppsala, Sweden. ✉e-mail: anna-sundborger-lunna@icm.uu.se

Insight into the structural basis of BAR-mediated membrane remodeling is limited to either static crystal structures of soluble (and often truncated) proteins[2,7,10,46,47], solution NMR structures of small individual domains[46,48], or low-resolution cryo-EM maps of helical scaffolds assembled on tubulated liposomes[16,49,50]. Furthermore, available crystallographic maps of N-BAR proteins show poor density for amphipathic motifs. This is likely due to the regions assuming helical conformation only when inserted into membranes[33]. For example, NMR analysis of membrane-bound BIN1 H0 reveals that the N-terminal end is highly flexible, but that the rest adapts a helical conformation[48]. One available N-BAR crystal structure (BIN2) includes the majority of H0[51]. Though, the position and orientation of H0 in that model may be called into question as H0 is locked between the BAR domains of symmetry mates in the unit cell. Since the structure contains a large fragment of an H0, it appears to have greatly influenced predictions for other N-BARs, as all AlphaFold models of N-BARs show H0 in a similar position and orientation[52,53].

Here we present the first single-particle cryo-EM structure of a membrane-bound BAR protein that reaches near-atomic resolution. This allows us to accurately determine the position and variable conformations of amphipathic regions, something that has severely limited previous studies of BAR proteins. Our EM structure of the endophilin B1 lipoprotein complex consists of six endophilin B1 dimers bound in two distinct conformations to a single artificial lipid platform. CryoDRGN analysis reveals multiple populations with different stoichiometries, as well as conformational variations of the amphipathic helices and BAR domain of endophilin B1. Structural analyses of full-length endophilin B1 in solution with cryo-EM and small-angle x-ray scattering (SAXS) show that the protein is highly flexible. A truncated form of endophilin B1, lacking a SH3 domain has a different shape in solution compared to the full-length protein. These results strengthen our hypothesis that, in solution, the SH3 domain occludes the concave side of the BAR domain, which contributes to autoinhibition. We also present evidence that endophilin B1 permeabilizes liposomes. The phenomenon is dependent on a negative surface charge and is enhanced by the presence of cardiolipin. Together, our results present a novel model for BAR protein scaffold assembly and propose a critical role for endophilin B1 in the permeabilization of the outer-mitochondrial membrane during programmed cell death.

## Results

### Endophilin B1 binds to cardiolipin-containing nanodiscs
To understand how endophilin B1 promotes diverse membrane remodeling events in the cell, we wanted to reveal the interaction between endophilin B1 and membranes. Our previous studies show that endophilin B1 organizes into helical polymers on cardiolipin-enriched liposomes in distinct modes[25]. We proposed that these diverse organizations were regulated by coordinated H0–H1i association with the membrane. However, limited resolution prevented further insight into the exact nature of these interactions. Therefore, to probe the ability of endophilin B1 amphipathic motifs to drive distinct modes of membrane association and subsequent remodeling, we used the limiting membrane supports of lipid nanodiscs[54–57]. Nanodiscs have been used extensively to resolve structures of integral membrane proteins[58], but only on one occasion to study the structure of a peripheral membrane protein[59]. We reasoned that these lipid bilayer scaffolds are big enough to allow endophilin B1 binding, but too small to allow polymerization into helical scaffolds.

We generated nanodiscs consisting of membrane scaffolding protein MSP2N2 and lipids with negatively charged headgroups, including cardiolipin (90% DOPS, 10% 14:0 cardiolipin). Nanodiscs incubated with endophilin B1 show a substantial shift in molecular weight towards larger assemblies according to size-exclusion chromatography (SEC; Fig. 1a). Incubating nanodiscs with endophilin B1 at different molar ratios (1:2 and 1:10; MSP2N2:Endophilin B1) results in significantly different elution profiles (Supplementary Fig. 1a). Endophilin B1 decoration of nanodiscs was confirmed by negative-stain EM (Supplementary Fig. 1b). The sample containing more endophilin B1 (1:10) corresponds to a larger particle size

than the sample with less endophilin B1 (1:2; Supplementary Fig. 1a). This indicates that higher ratios of endophilin B1 results in nanodiscs with more endophilin B1 decoration.

### Endophilin B1-lipoprotein complexes have different stoichiometries
Cryo-EM data of the 1:2 molar ratio sample were split into 2 classes. Most particles appeared to consist of empty MSP2N2 nanodiscs (Supplementary Fig. 1c) with a smaller subset of lipoprotein complexes decorated with one endophilin B1 dimer (Supplementary Fig. 1d). Interestingly, the diameter of decorated lipoprotein complexes is roughly 2 nm smaller than that of undecorated nanodiscs. Similarly, lipoprotein complexes decorated with six dimers in the 1:10 sample are smaller than undecorated nanodiscs (Supplementary Fig. 1e).

### Amphipathic regions of endophilin B1 anchor to the membrane
The high-resolution data set of endophilin B1-decorated bicelles (1:10) contains both conformational and compositional heterogeneity, despite eluting as a single SEC peak (Fig. 1a, b). Electron density maps consisting of >5 membrane-bound endophilin B1 dimers were improved iteratively during several rounds of multi-class heterogeneous ab-initio reconstruction followed by heterogeneous refinement in cryoSPARC (for the complete workflow, see Supplementary Fig. 2)[60]. The final volume consisted of six endophilin B1 dimers and had a nominal resolution of 3.88 Å (Supplementary Fig. 3a, Table 1). Strong density could be observed for four dimers and slightly weaker density for two additional dimers. The volume contains a membrane bilayer density with anchored amphipathic helices (Fig. 1b). Interestingly, at higher contour levels the density corresponding to the membrane bilayer disappears, and where we expect to observe density corresponding to MSP2N2 scaffolds, there is none (Fig. 1c). Instead, we observe density that belongs to endophilin B1 amphipathic helices. This, together with the smaller observed diameter compared to MSP2N2 nanodiscs, is evidence that the particles in the cryo-EM reconstruction are not endophilin-decorated nanodiscs but rather lipoprotein complexes consisting of endophilin B1 decorated bicelles.

The local resolution of the BAR domains is higher than that of the inserted amphipathic helices, suggesting the protein is flexible at the protein-lipid interface (Fig. 1c). The highest local resolution is found at the center of the BAR domains, at the dimerization interface, however, the distal ends of the BAR domains appear flexible.

### Endophilin B1 amphipathic regions undergo major conformational changes upon membrane binding
Together, six dimers form a cage ("mini scaffold") around a patch of the bilayer, where each dimer makes direct contact with neighboring dimers through their respective H0 and H1i motifs (Fig. 1c). Dimers bound to the surface of the bilayer and dimers that straddle the edges of the bicelle have distinct amphipathic helix conformations (Fig. 2a and b). These different classes of dimers will be referred to as "center" and "side" dimers, respectively.

In center dimers, H1i is predominantly disordered. At lower contour levels the disordered region appears close to the BAR domain, brushing the membrane surface, but not inserted. H0s are oriented anti-parallel in relation to each other. In side dimers, both helical and loop regions of H1i have visible density, and H1i is inserted into the membrane. Interestingly, H0 are oriented parallel to each other. Locally refining the classes individually resulted in reconstructions of 3.45 Å (center dimer; Supplementary Fig. 3b) and 3.60 Å (side dimer; Supplementary Fig. 3c), respectively. These two maps were used to build atomic models (Fig. 2b, Table 1). Focused refinements of individual amphipathic motifs did not increase the resolution at the membrane interface, most likely due to their size and mobility at the surface.

The atomic models consist of residues Leu11–Leu252. Residues Met1-Lys10 of H0 were omitted as weak density indicates that the N-terminus is

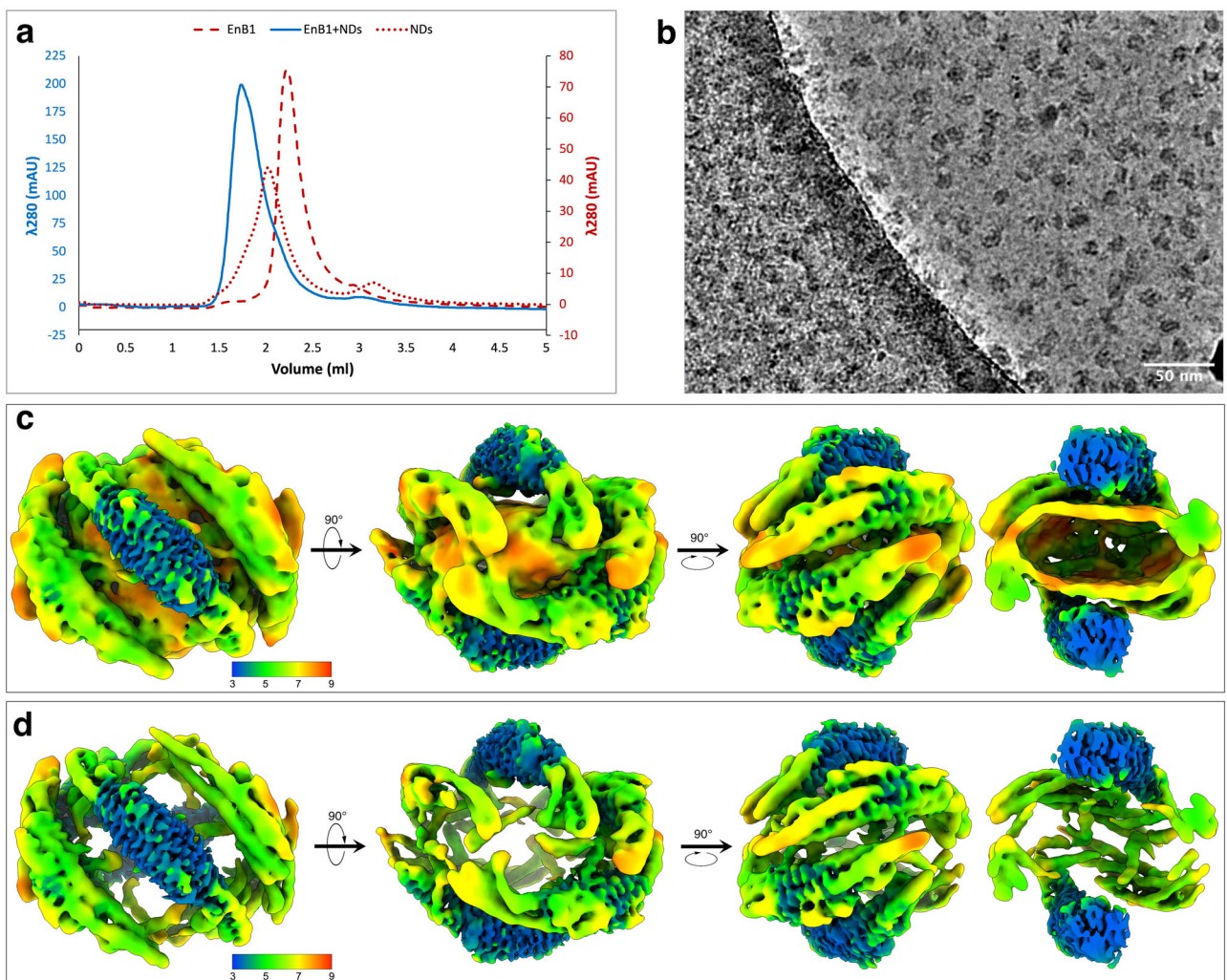

**Fig. 1 | Cryo-EM structure of endophilin B1-decorated bicelles. a** Representative SEC runs of endophilin B1, MSP2N2 nanodiscs (NDs) and nanodiscs incubated with endophilin B1 at a molar ratio of 1:10 (MSP2N2:Endophilin B1). **b** Raw micrograph from high-resolution endophilin B1-decorated bicelles data collection. **c** and **d** 3D reconstruction of endophilin B1-decorated bicelles at different contour levels showing the local resolution.

flexible while inserted into the membrane (Fig. 2e). This portion of the N-terminus has a net charge of +1. Residues L11–L30 assume a helical conformation with a net charge of −1 (Table 2). As H1i is disordered in the center dimer class, residues Glu76-Ile96 were deleted from the model (Supplementary Fig. 3d). However, there is clear density corresponding to the whole H1i including the loop connecting H0 to the BAR domain in the side dimer reconstruction (Supplementary Fig. 3e).

**The H2–H3 hinge region is flexible when membrane-bound**

The BAR domain H2 can be split into two segments. The portion that contributes to dimer formation is more rigid, which is reflected in higher local resolution in the EM maps for that section (Supplementary Fig. 3f and g). The portion that extends away from the dimer interface has a significantly weaker density, indicating that there is conformational heterogeneity present. The residues between these segments form a hinge that moves in concert with the hinge region that is also present in H3. The key residues that appear to facilitate movement are two glycines, Gly153 and Gly215, that nick their respective helices, splitting them into rigid and more flexible sections (Supplementary Fig. 3h). Docking the models into the original EM map (Fig. 2c) suggests that inter-dimer contact sites consist of hydrophobic interactions between leucines and valines (Leu19, 30, 83 and Val23) (Fig. 2d).

**Neural network analysis reveals side-to-side assembly and distortion of bicelle shape**

The workflow described above yielded two 3D reconstructions of near-atomic resolution that were used to build atomic models. However, we were also interested in assemblies with different stoichiometries that had been filtered away during the refinement process. To untangle the compositional and conformational heterogeneity present in the sample, we used cryoDRGN[61,62]. Two groups of particles, one which resulted in a high-resolution reconstruction and another, further upstream in the cryoSPARC workflow (maps * and †, respectively, in Supplementary Fig. 2), were exported to cryoDRGN. Training results for the high-resolution reconstruction reveal that a shift in H0 density is coupled with conformational changes in the side dimer BAR domain (Movie 1). Principle component (PC) analysis (Fig. 3a) shows that the shape of the bicelle changes and becomes more elliptical as the distance between the distal tips of side dimers on opposite sides of the platform increases by ~1–2 nm (Movie 2, Fig. 3). As the bicelle becomes more elliptical, the center dimers H0s move closer together, and the membrane density becomes more distinct. CryoDRGN heterogeneous ab-initio reconstructions of the larger group of particles effectively sorted particles with different stoichiometries (Fig. 4a, Movie 3). 3D reconstructions were generated for bicelle classes with 3–6 endophilin B1 dimers, at nominal resolutions between 6.40 and 9.59 Å (Fig. 4b–k).

**Table 1 | Cryo-EM data collection, refinement, and validation statistics**

| | #1 Endophilin B1 bound to MSP2N2 nanodiscs (consensus map) (EMDB-50981) (PDB 9G2R) | #2 Endophilin B1 bound to MSP2N2 nanodiscs (center dimer focused map) (EMDB-50984) (PDB 9G2U) | #2 Endophilin B1 bound to MSP2N2 nanodiscs (side dimer focused map) (EMDB-50986) (PDB 9G2W) |
|---|---|---|---|
| *Data collection and processing* | | | |
| Magnification | 130,000 | 130,000 | 130,000 |
| Voltage (kV) | 300 | 300 | 300 |
| Electron exposure (e–/Å²) | 40 | 40 | 40 |
| Defocus range (μm) | −1.2 to −2.2 | −1.2 to −2.2 | -1.2 to -2.2 |
| Pixel size (Å) | 0.664 | 0.664 | 0.664 |
| Symmetry imposed | C1 | C1 | C1 |
| Initial particle images (no.) | 5,026,978 | 5,026,978 | 5,026,978 |
| Final particle images (no.) | 273,120 | 273,120 | 273,120 |
| Map resolution (Å)<br>FSC threshold | 3.88<br>0.143 | 3.45<br>0.143 | 3.60<br>0.143 |
| Map resolution range (Å) | 2.98–30.00 | 2.98–30.00 | 2.98 to 30.00 |
| *Refinement* | | | |
| Initial model used (AlphaFold code) | AF-Q9Y371-F1-model_v4 | AF-Q9Y371-F1-model_v4 | AF-Q9Y371-F1-model_v4 |
| Model resolution (Å)<br>FSC threshold | 3.9<br>0.143 | 3.4<br>0.143 | 3.5<br>0.143 |
| Model resolution range (Å) | 3.26–8.96 | 2.98–7.38 | 3.00 to 7.02 |
| Map sharpening *B* factor (Å²) | 108.8 | 91.5 | 115.4 |
| Model composition | | | |
| Non-hydrogen atoms<br>Protein residues<br>Ligands | 22,620<br>28,24<br>0 | 3634<br>456<br>0 | 3838<br>478<br>0 |
| *B* factors (Å²) | | | |
| Protein<br>Ligand | 93.50/273.63/156.42<br>N/A | 93.50/201.89/141.75<br>N/A | 105.94/273.63/163.37<br>N/A |
| R.m.s. deviations | | | |
| Bond lengths (Å)<br>Bond angles (°) | 0.011 (0)<br>1.242 (0) | 0.012 (0)<br>1.328 (0) | 0.010 (0)<br>1.200 (0) |
| Validation | | | |
| MolProbity score<br>Clashscore<br>Poor rotamers (%) | 1.48<br>8.96<br>0.58 | 1.52<br>9.87<br>0.26 | 1.58<br>11.68<br>0.73 |
| Ramachandran plot | | | |
| Favored (%)<br>Allowed (%)<br>Disallowed (%) | 98.28<br>1.72<br>0.00 | 98.66<br>1.34<br>0.00 | 98.09<br>1.91<br>0.00 |

These reveal how endophilin B1 mini scaffolds assemble side-to-side on bicelles.

## The linker-SH3 domain region is flexible when endophilin B1 is membrane-bound

We have previously shown that the SH3 domain negatively regulates the ability of endophilin B1 to mediate membrane tubulation, suggesting it plays an important role in the regulation of endophilin B1 membrane activity[25]. In solution, the SH3 domain associates with H0. This observation led us to propose that, in solution, the SH3 domain is fixed in position near the concave surface of the BAR domain by H0 and that H0 association with membranes regulates its conformation and subsequently, the ability of endophilin B1 to cause membrane curvature. Previous studies of endophilin A1 propose that the SH3 domain may bind opposite ends of the BAR dimer[23,63]. Interestingly, no density corresponding to the SH3 domain-linker region is visible in any of our maps, either near the concave surface or proximal to the lateral part of the BAR domain. This indicates that this region is highly flexible when endophilin B1 is membrane-bound. To determine whether membrane binding regulates the organization of this flexible region, we attempted to determine the structure of the cytosolic, i.e., the soluble state of endophilin B1 using cryo-EM. However, vitrified grids consistently had poor ice quality and particle distribution, with most adhering to the carbon, to the edge of holes or clustering in discrete patches (Supplementary Fig. 4a and b). Low concentration of protein (5 μM) yielded slightly better ice, which allowed the collection of a small data set and generation of 2D class averages (Supplementary Fig. 4c). Ultimately, no useful 3D reconstruction could be obtained due to preferred particle orientation.

Small-angle X-ray scattering (SAXS) data was collected using soluble endophilin B1 at three different concentrations (4, 8, and 16 μM; Table 3, Supplementary Figs. 5a–c and 6a). We observe differences in the radius of gyration ($R_G$) and the polydispersity of the protein at different

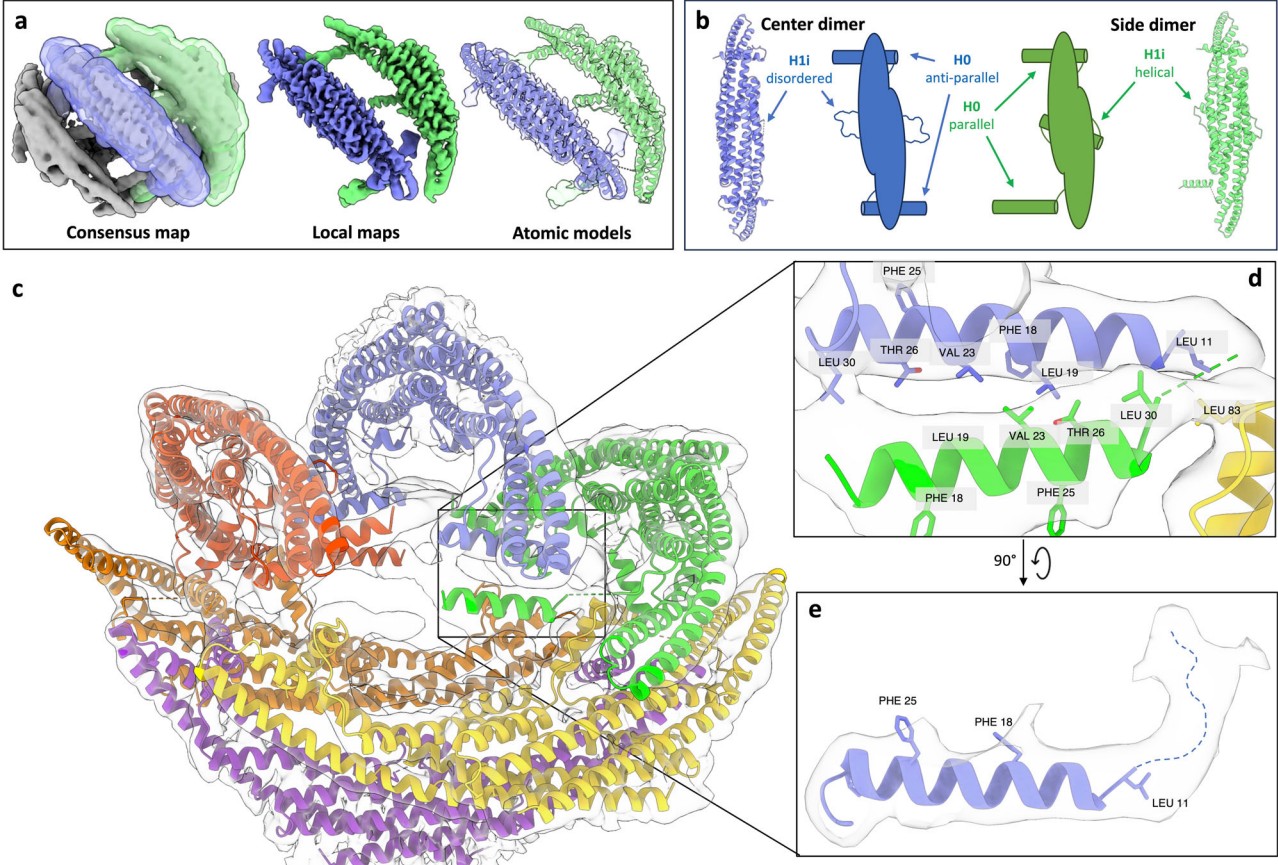

**Fig. 2 | Focused refinement and atomic models of membrane-bound endophilin B1. a** Left: Electron density map of endophilin B1 on a nanodisc with masks for one center dimer and one side dimer. Middle: Resulting local refinement maps of the center and side dimers. Right: Atomic models docked into both density maps. **b** Top views of the atomic models and cartoon representations highlighting the different conformations of the amphipathic helices. **c** Models for all dimers docked into the original density map. **d** Interactions between multiple amphipathic motifs. **e** The N-terminus of H0, is flexible in the structure. The dotted line represents the flexible N-terminus in the unmodeled part of the H0 electron density.

### Table 2 | Multiple sequence alignment of endophilin H0 helices

| H0 | Sequence | |
|---|---|---|
| Endophilin A1 | ---------MSVAGLKKQFHKATQKVSEKVG | 22 |
| Endophilin A2 | ---------MSVAGLKKQFYKASQLVSEKVG | 22 |
| Endophilin A3 | ---------MSVAGLKKQFHKASQLFSEKIS | 22 |
| Endophilin B1 | MNIMDFNVKK**LAADAGTFLSRAVQFTEEKLG** | 31 |
| Endophilin B2 | ---MDFNMKKLASDAGIFFTRAVQFTEEKFG | 28 |

Endophilin B1 has a longer H0 than other endophilins. Residues belonging to the helical portion of endophilin B1 H0 are shown in bold.

concentrations (Supplementary Fig. 6b). At higher concentrations (8–16 μM), endophilin B1 is primarily dimeric and partly disordered, whereas at low concentrations (4 μM), it is monomeric and mostly disordered (Supplementary Fig. 6c–e). Analysis of SAXS data of truncated endophilin B1 lacking the SH3 domain (endophilin B1_ΔSH3; Table 3, Supplementary Figs. 5d and 7b) indicates that the mutated protein is more globular than the wild-type form (Supplementary Fig. 7c–f). We observe a decrease in $R_G$ compared to the wild type. The SAXS ab-initio reconstructions of wild-type endophilin B1 and endophilin B1_ΔSH3 show BAR domains with distinct shapes. The reconstructions are similar in length; however, the wild-type reconstruction has added density on the concave side of the BAR domain (Supplementary Fig. 7f). Endophilin B1_ΔSH3 and full-length endophilin B1 have similar elution volumes in SEC despite a predicted ~12 kDa difference in MW (Supplementary Fig. 7a).

## Endophilin B1 permeabilizes membrane vesicles

It has been previously shown that insertion of amphipathic regions can cause membrane permeabilization[64–68]. To determine whether endophilin B1-mediated membrane disruption drives membrane permeabilization, we added endophilin B1 to liposome with encapsulated quenched calcein (Table 4). We find that the addition of endophilin B1 to these membrane vesicles with a net negative surface charge leads to a significant increase in calcein fluorescence as a result of membrane permeabilization and de-quenching (Fig. 5; Supplementary Data 1). Interestingly, the ability of endophilin B1 to promote membrane permeabilization is dependent on the lipid composition. Endophilin B1 shows low permeabilization activity when added to liposomes with a net neutral charge (Mix 1) and significantly higher activity when added to net negatively charged liposomes (Mix 2) (Fig. 5a). We observed more permeabilization when vesicles were prepared with the addition of phospholipids that contribute to membrane packing defects (Mix 3)[69,70]. This effect is further increased when cholesterol is removed (Mix 4). Robust permeabilization of vesicles was observed when a higher concentration of endophilin B1 (500 nM vs. 5 μM, $p < 0.01$) was added to vesicles with 20% cardiolipin (Mix 5) (Fig. 5b). The most efficient permeabilization is observed at 37 °C.

## Discussions
Endophilin B1 binds to nanodiscs with a stoichiometry proportional to the initial molar ratio between endophilin B1 and MSP2N2. The 1:10 molar ratio cryo-EM data set contained substantial compositional and

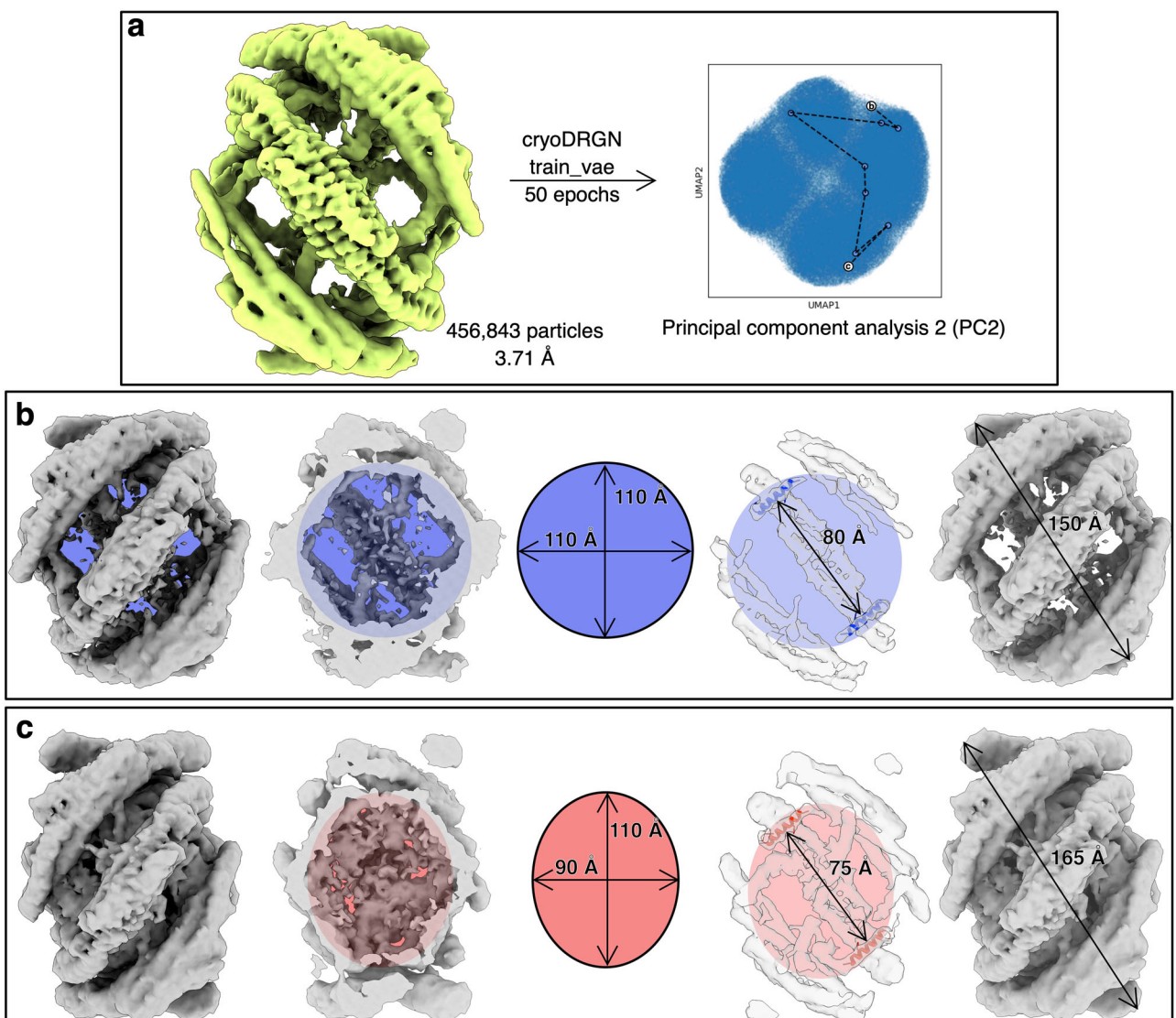

**Fig. 3 | CryoDRGN predicts that endophilin B1 distorts the shape and affects the membrane fluidity of lipid bilayers. a** 3D reconstruction of particles used for analysis with cryoDRGN train_vae and the resulting UMAP distribution showing traversal along principal component 2 (PC2 traversal can be seen in Movie 3). Volumes generated by cryoDRGN representing the minima (**b**) and maxima (**c**) of PC2. CryoDRGN predicts a more intense signal from the lipid bilayer in (**c**) compared to (**b**).

conformational heterogeneity. Through saturation of the available lipid surface area during sample preparation and extensive cryo-EM data processing of a large data set, a near-atomic resolution reconstruction could be produced for particles with 6 endophilin B1 dimers bound per bicelle. These structures are the highest-resolution EM structures of a BAR protein to date. Another benefit to this approach for studying peripheral membrane proteins is that it can capture the shape and orientation of membrane-bound amphipathic regions.

Initially, we suspected that the amphipathic helices of endophilin B1 might cling to MSP2N2. Surprisingly, we found no density for MSP2N2 in our maps, despite its presence in the sample confirmed by Western Blot analysis (Supplementary Fig. 9). Only amphipathic regions of endophilin B1 are present in our final map. We speculate that destabilization of the nanodisc lipid bilayer by endophilin B1 causes the displacement of MSP2N2. It is replaced by the amphipathic regions of endophilin B1, which stabilize the bilayer. Superimposing the NMR structure of MSP1D1 onto our structure shows how the amphipathic helices of endophilin B1 would clash with that of MSP2N2 (Supplementary Fig. 10)[71]. There are multiple reports of proteins and peptides (in addition to MSPs) that can form

discoidal lipoprotein particles[72–75]. Perhaps isolated endophilin H0s could represent another method of creating nanodisc-like lipoprotein particles similar to the Salipro system[73].

Dimers could be divided into two categories based on where they bound to bicelles; either on the flatter faces (center) or on the more curved edges of bicelles (side). Focused refinement revealed that the major differences between these two dimers are the conformations of their respective amphipathic motifs. H0 is membrane-bound in both dimer categories, but H1i is not, which indicates that H0 is responsible for initial membrane binding and that H1i insertion occurs later.

In the consensus structure, we observe that intermolecular interactions occur on the membrane surface through amphipathic motifs (Fig. 2c). These motifs all bind to or near the bicelle edge, where local curvature is the highest. H0s organize anti-parallel to each other. The loop that links H0 to the BAR domain is flexible enough to accommodate H0 twisting ~180°, allowing side-to-side assembly. Diverse orientations of H0 positioning could explain why endophilin B1 (and other endophilins) produce heterogeneous helical scaffolds[23,25,49], as there are multiple plausible ways it could oligomerize to form scaffolds (Fig. 6a).

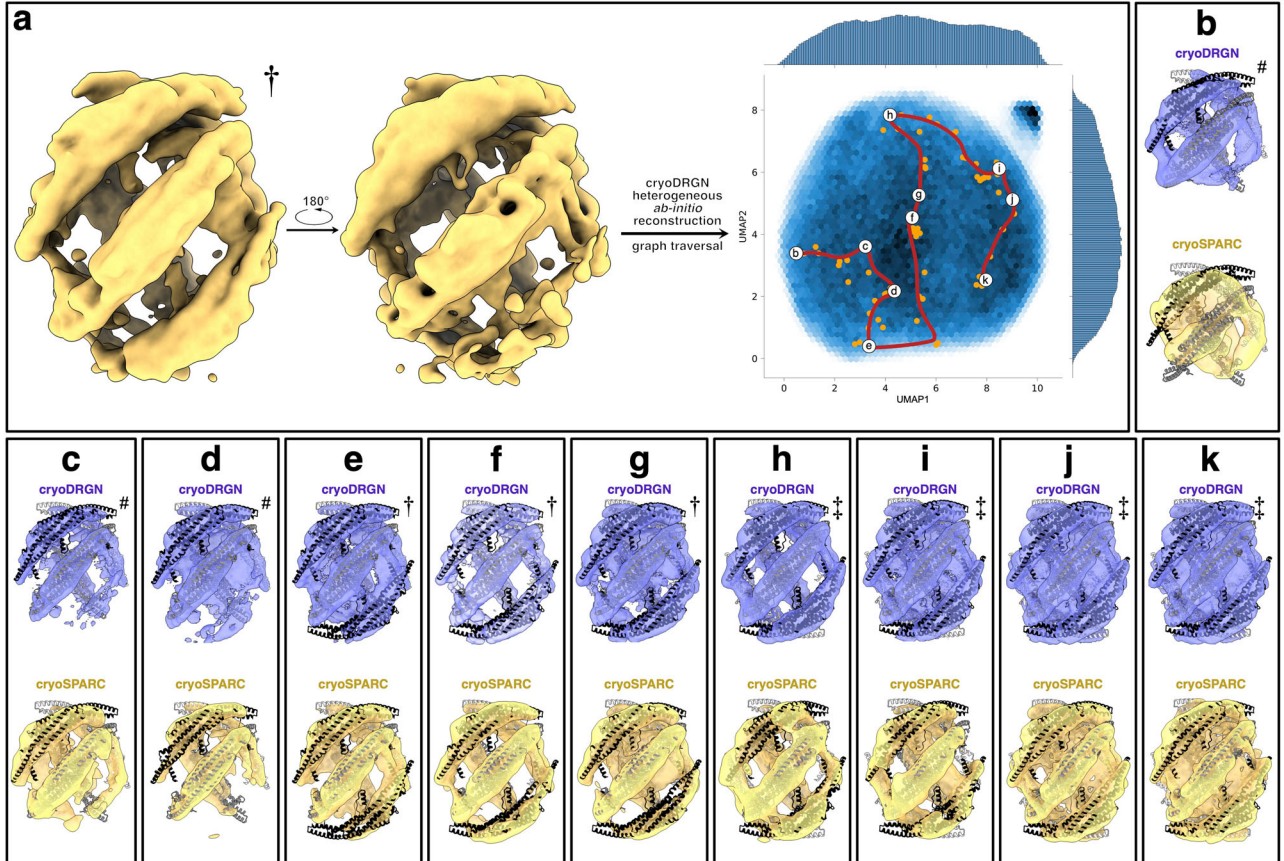

**Fig. 4 | CryoDRGN reveals side-to-side assembly of endophilin B1 on lipid bicelles. a** 3D reconstruction of particles (758,658 particles; 6.22 Å) used in cryoDRGN heterogeneous ab-initio reconstruction and the resulting UMAP distribution showing locations of different k-means clustered classes and the path of the graph traversal. Symbols mark the different stoichiometries present in each class of particles: (#) four, (†) five, or (‡) six dimers bound per nanodisc. **b** k-means cluster 0;

72,644 particles; 9.59 Å, **c** k-means cluster 1; 81,574 particles; 7.42 Å, **d** k-means cluster 2; 72,337 particles; 8.15 Å, **e** k-means cluster 3; 80,745 particles; 7.53 Å, **f** k-means cluster 4; 79,658 particles; 8.03 Å, **g** k-means cluster 5; 83,754 particles; 7.85 Å, **h** k-means cluster 6; 73,407 particles; 8.65 Å, **i** k-means cluster 7; 33,465 particles; 8.54 Å, **j** k-means cluster 8; 96,256 particles; 6.46 Å, **k** k-means cluster 9; 84,818 particles; 6.40 Å.

There was unexpected variability in the conformations of helices 2 and 3 at the tips of the BAR domain. CryoDRGN results show that the BAR domain is flexible and can assume and induce different levels of curvature. This is also apparent when the two atomic models are compared. This is mainly due to kinks in helices 2 and 3 that accommodate the hinge movement. Kinks, and therefore possible hinge movement, are present in other BAR proteins, such as sorting nexins, FCHo, and syndapin[4]. In the H2 and H3 kinks of endophilin B1, there are two glycines, Gly153 and Gly215. These residues are conserved in endophilins B1/2, but not in the endophilin A subfamily, suggesting that endophilin B1 and B2 can adapt to and induce a wider range of curvature than endophilins in the A subfamily. This further supports our notion that endophilin B1 is uniquely capable of promoting diverse membrane remodeling events.

Our previous data suggests that the SH3 domain autoinhibits the membrane remodeling activity of endophilin B1[25]. Interestingly, we observe no SH3 domain density in our maps of membrane-associated endophilin B1 (Fig. 1). Furthermore, attempts to determine the binding location of the SH3 domain in the soluble form of the protein were inconclusive (Supplementary Fig. 4). Soluble endophilin B1 is highly flexible (Supplementary Fig. 6c), indicating that the linker-SH3 domain region adopts more than one conformation in solution. However, wild-type endophilin B1 does contain additional density on the concave side of the protein, between the H0s (Supplementary Fig. 6e). We speculate that this density constitutes the SH3-linker region and that the intramolecular interactions are dynamic. Interestingly, the removal of the SH3 domain renders endophilin B1 less flexible (Supplementary Fig. 7d). This indicates that these intramolecular

interactions are more likely without the SH3 domain. We propose that the linker is responsible for mediating intramolecular interactions and that these interactions are weakened by the presence of the SH3 domain.

Endophilin B1 is monomeric in solution, as evidenced by 2D classes, where only three BAR domain helices can be observed (Supplementary Fig. 4), and SAXS analysis (Supplementary Fig. 6b). Furthermore, monomeric endophilin B1 is more polydisperse than its dimeric form. This is corroborated by observations that dimerization decreases the flexibility of endophilin B1. This may be due to the stabilization of the alpha helices that constitute the dimer interface upon assembly. It is also possible that intramolecular interactions involving the linker-SH3 domain are weaker in the monomeric form, which would increase polydispersity.

We observe stronger density for the lipid bilayer when the BAR domain is in a "flexed" position, i.e. the H0s of individual endophilin dimers are closer together (Fig. 4). This indicates that endophilin B1 mini scaffold assembly greatly disturbs the membrane bilayer. Interestingly, we also observe side-to-side assembly of endophilin B1 mini scaffold on large cardiolipin-containing membrane liposome (Supplementary Fig. 11), suggesting that this form of endophilin B1 assembly occurs even when space is not limiting (i.e. on a nanodisc). Both endophilin scaffolds and isolated H0s limit lipid diffusion, which has been suggested to cause vesiculation[76,77]. It was previously reported that endophilin B1 induces minor leakage of fluorescein isothiocyanate-labeled dextran-loaded liposomes[27]. In this study, we show that endophilin B1 effectively permeabilizes mitochondria-like liposomes and that this activity is dependent on the membrane lipid composition, specifically the presence of negatively charged lipids (DOPS

**Table 3 | SAXS data collection and refinement statistics**

**(a) Sample details**

| | Endophilin B1_wt 0.69 mg/ml | Endophilin B1_wt 0.34 mg/ml | Endophilin B1_wt 0.14 mg/ml | Endophilin B1_ΔSH3 |
|---|---|---|---|---|
| Description of sequence | Endophilin-B1 (Q9Y371) from *Homo sapiens* | | | |
| | Full-length | | | Truncated (1–306a.a.) |
| Extinction coefficient $\varepsilon$ (M$^{-1}$ cm$^{-1}$) | 29,130 (280 nm) | | | 19,160 (280 nm) |
| Partial specific volume (cm$^3$/g) | 0.72865 | | | 0.60219 |
| Mean solute and solvent SLD (10$^{-6}$ Å$^{-2}$)$^2$ | 12.475/9.465 | | | 15.094/9.465 |
| Mean scattering contrast (10$^{-6}$ Å$^{-2}$)$^2$ | 3.010 | | | 5.629 |
| Molecular mass (Da)$^2$ | 40,794 | | | 34,219 |
| Sample concentration (mg/ml) | 0.69 | 0.34 | 0.14 | 2.5 |
| Solvent composition | 150 mM NaCl, 20 mM HEPES, 1 mM TCEP, 0.5 mM DTT, pH 8.1 | | | |

**(b) SAS data collection parameters**

| | |
|---|---|
| Instrument | ESRF BM29 |
| Wavelength (Å) | 0.9918 |
| Beam geometry | Size: 700 × 700 µm$^2$; Sample-to-detector distance: 2.827 m |
| Sample configuration | 1.0 mm-diameter quartz capillary |
| $q$-measurement range (Å$^{-1}$) | 0.0025–0.6 |
| Absolute scaling method | Comparison with scattering from pure H$_2$O |
| Basis for normalization to constant counts | To transmitted intensity by direct beam counter |
| Exposure time | 15 s |
| Sample temperature (°C) | 4 |

**(c) Software employed for SAS data reduction, analysis and interpretation**

| | |
|---|---|
| SAS data averaging and subtraction | PRIMUS from ATSAS 3.2.1 |
| Calculation of $\varepsilon$ from sequence | ProtParam: https://web.expasy.org/protparam/ |
| Calculation of values from chemical composition | Peptide Property Calculator: http://biotools.nubic. northwestern.edu/proteincalc.html |
| Calculation of values from chemical composition | SLD calculator web: http://www.refcalc.appspot. com/sld |
| Guinier, $P(r)$ | GNOM from ATSAS |
| Atomic structure map modeling | DENSSWeb (v 1.7.0): https://denss.ccr. buffalo.edu |
| Molecular graphics | ChimeraX-1.6.1 |

**(d) Structural parameters**

| Guinier analysis | Endophilin B1_wt 0.69 mg/ml | Endophilin B1_wt 0.34 mg/ml | Endophilin B1_wt 0.14 mg/ml | Endophilin B1_ΔSH3 |
|---|---|---|---|---|
| $I(0)$ (cm$^{-1}$) | 73.13 ± 0.28 | 66.06 ± 0.57 | 43.00 ± 0.75 | 31.25 ± 0.07 |
| $R_G$ (Å) | 50.59 ± 2.34 | 49.67 ± 6.09 | 36.88 ± 9.56 | 46.18 ± 2.62 |
| $q$ $R_G$-range | 0.5748–1.4567 | 0.4702–1.3158 | 0.4571–1.3367 | 0.4632–1.3461 |
| *P(r) analysis* | | | | |
| $I(0)$ (cm$^{-1}$) | 72.26 ± 0.23 | 65.95 ± 0.36 | 45.16 ± 0.65 | 31.42 ± 0.06 |
| $R_G$ (Å) | 51.05 ± 1.70 | 51.04 ± 3.60 | 42.70 ± 6.5 | 47.60 ± 1.1 |
| $d_{max}$ (Å) | 160 | 160 | 153 | 152 |
| $q$-range (Å$^{-1}$) | 0.0114–0.3009 | 0.0095–0.3009 | 0.0124–0.3009 | 0.0085-0.2390 |
| Total quality estimate (GNOM) | 0.8429 | 0.845 | 0.6489 | 0.552 |

**(e) Atomistic modeling**

| | Endophilin B1_wt 0.69 mg/ml | Endophilin B1_wt 0.34 mg/ml | Endophilin B1_wt 0.14 mg/ml | Endophilin B1_ΔSH3 |
|---|---|---|---|---|
| Method | DENSS 1.7.0 | | | |
| $q$-range for fitting | 0.0114–0.3009 | 0.0095–0.3009 | 0.0124–0.3009 | 0.0085–0.2390 |
| Correlation score | 0.933 | 0.934 | 0.947 | 0.969 |

**(f) Data and model deposition IDs**

| | Endophilin B1_wt 0.69 mg/ml | Endophilin B1_wt 0.34 mg/ml | Endophilin B1_wt 0.14 mg/ml | Endophilin B1_ΔSH3 |
|---|---|---|---|---|
| | SASDVR4 | SASDVS4 | SASDVT4 | SASDVU4 |

**Table 4 | Permeabilization assay using endophilin B1 and liposomes of different lipid compositions (lipid ratio shown as molar percentage) where _n_ is the number of replicates**

| Name | Lipid composition | Protein conc. (µM) | Temperature (°C) | n | Calcein release (%) |
|---|---|---|---|---|---|
| Mix 1 | 100% DOPC | 0.5 | 25 | 18 | 0.68 ± 0.14 |
| Mix 2 | 55% DOPS, 40% DOPE, 5% Cholesterol | 0.5 | 25 | 18 | 6.9 ± 0.3 |
| Mix 3 | 50% DOPS, 40% DOPE, 5% Cholesterol, 5% 18:1 cardiolipin | 0.5 | 25 | 18 | 10 ± 1 |
| Mix 4 | 55% DOPS, 40% DOPE, 5% 18:1 cardiolipin | 0.5 | 25 | 18 | 11 ± 1 |
| Mix 5 | 40% DOPS, 40% DOPE, 20% 18:1 cardiolipin | 0.5 | 25 | 9 | 11 ± 1 |
| | | 0.5 | 37 | 9 | 27 ± 2 |
| | | 5 | 25 | 6 | 65 ± 6 |
| | | 5 | 37 | 6 | 96 ± 3 |

Results are shown as the mean ± standard deviation of the mean.

**Fig. 5 | Endophilin B1-mediated liposome permeabilization is dependent on membrane lipid composition, protein concentration, and temperature.** Strip plots showing % calcein release from liposomes with different lipid compositions (Table 4) after incubation with endophilin B1 (**a**) and % calcein release from liposomes containing lipid mix 5 after incubation with increasing concentrations of endophilin B1 at room-temperature and physiological temperature (**b**). Data represent sample distribution and mean, $n = 18$ (**a**) and $\geq 6$) (**b**), ***$p < 0.01$.

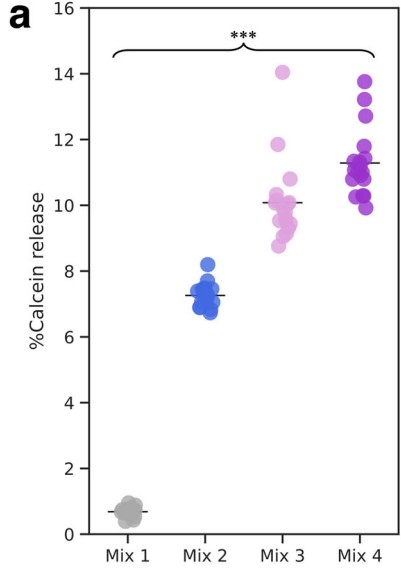

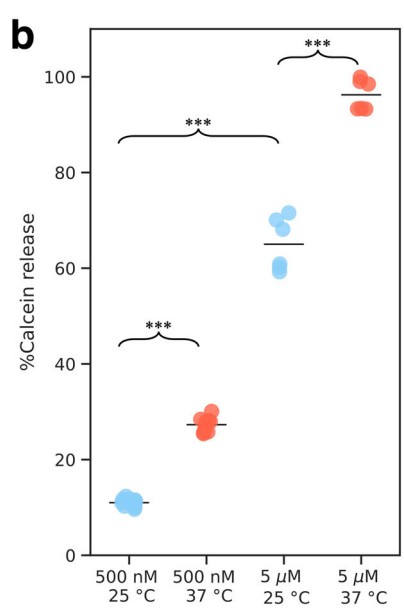

and cardiolipin), temperature, and protein concentration (Fig. 5). Moreover, we present the structural basis for endophilin B1-mediated membrane remodeling, revealing how H0 and H1i insertion and interactions cause destabilization.

The role of BAR protein helical assembly in the regulation of membrane remodeling is well established[3,4,6]. A well-studied example is that of endophilin A1 during endocytosis. When endophilin A1 binds to PIP$_2$-rich sites on the inner leaflet of the plasma membrane and oligomerizes, it stabilizes positive curvature by the formation of a helical scaffold (Fig. 6b)[78,79]. The SH3 domain then recruits effector proteins, including dynamin 1, which promotes membrane scission and helical scaffold disassembly[22]. However, it has not been determined how endophilins interact with membranes rich in cardiolipin—a cone-shaped lipid that favors negative curvature—such as the OMM[69,80,81].

Following apoptotic stimuli and increased generation of reactive oxygen species, oxidized cardiolipin migrates to the OMM (Fig. 6c)[82–84], where it may act as a recruitment platform for Bax[85]. We propose that during apoptosis, endophilin B1 localizes to cardiolipin-rich areas on the OMM, where it stabilizes the negative curvature generated by cardiolipin clustering. As a result of the local membrane curvature, endophilin B1 assembly is limited to circular mini scaffolds, where dimers interact side-to-side instead of end-to-end. Further clustering of cardiolipin into pits is stabilized by the mini scaffolds limiting lipid diffusion. This creates areas of highly unstable membrane patches onto which Bax and other pro-apoptotic factors are recruited. Once they are inserted into the membrane and oligomerize, OMM permeabilization, cytochrome-c leakage, and ultimately, cell death

are inevitable. In summary, we show multiple features of the structural basis for endophilin B1 membrane disruption, which on a membrane rich in packing defects and negative charge, leads to permeabilization. We propose that endophilin B1 and cardiolipin coordinate to promote Bax-mediated cell death through an interplay of scaffold assembly, negative curvature, and chaotic insertion of amphipathic regions.

## Methods

### Expression and purification of endophilin-B1

His$_{12}$-SUMO-endophilin-B1, both full-length and truncated forms[25], were expressed recombinantly in _E. coli_ BL21(DE3) cells. Bacteria were grown in lysogeny broth (LB) and induced with 0.2 mM IPTG at OD$_{600}$ = 0.6, followed by overnight expression at 20 °C. Cells were harvested and resuspended in cell lysis buffer (20 mM Tris, 500 mM NaCl, 20 mM Imidazole, 0.5 mM TCEP, and 0.25% Triton X-100, pH 8.2). Before lysis, cOmplete™ Protease Inhibitor Cocktail (Sigma-Aldrich) and Dnase were added. Cells were lysed at 35 kPSI using a cell disruptor (Constant Systems), and the lysate was centrifuged at 30,600 × _g_ for 60 min. Clarified lysate was applied to the gravity column containing Ni-NTA affinity resin pre-equilibrated with binding buffer (20 mM Tris, 500 mM NaCl, 20 mM Imidazole, 0.5 mM TCEP, pH 8.2). The column was washed with wash buffer (20 mM Tris, 500 mM NaCl, 50 mM Imidazole, 0.5 mM TCEP, pH 8.2) and endophilin-B1 eluted with elution buffer (20 mM Tris, 500 mM NaCl, 500 mM Imidazole, 0.5 mM TCEP, pH 8.2). The eluted protein was dialyzed into cleavage buffer (20 mM Tris, 150 mM NaCl, 0.5 mM TCEP, pH 8.0) and SUMO (Ulp1) protease was added to dialyzed protein at 1:10 (w/w) for overnight

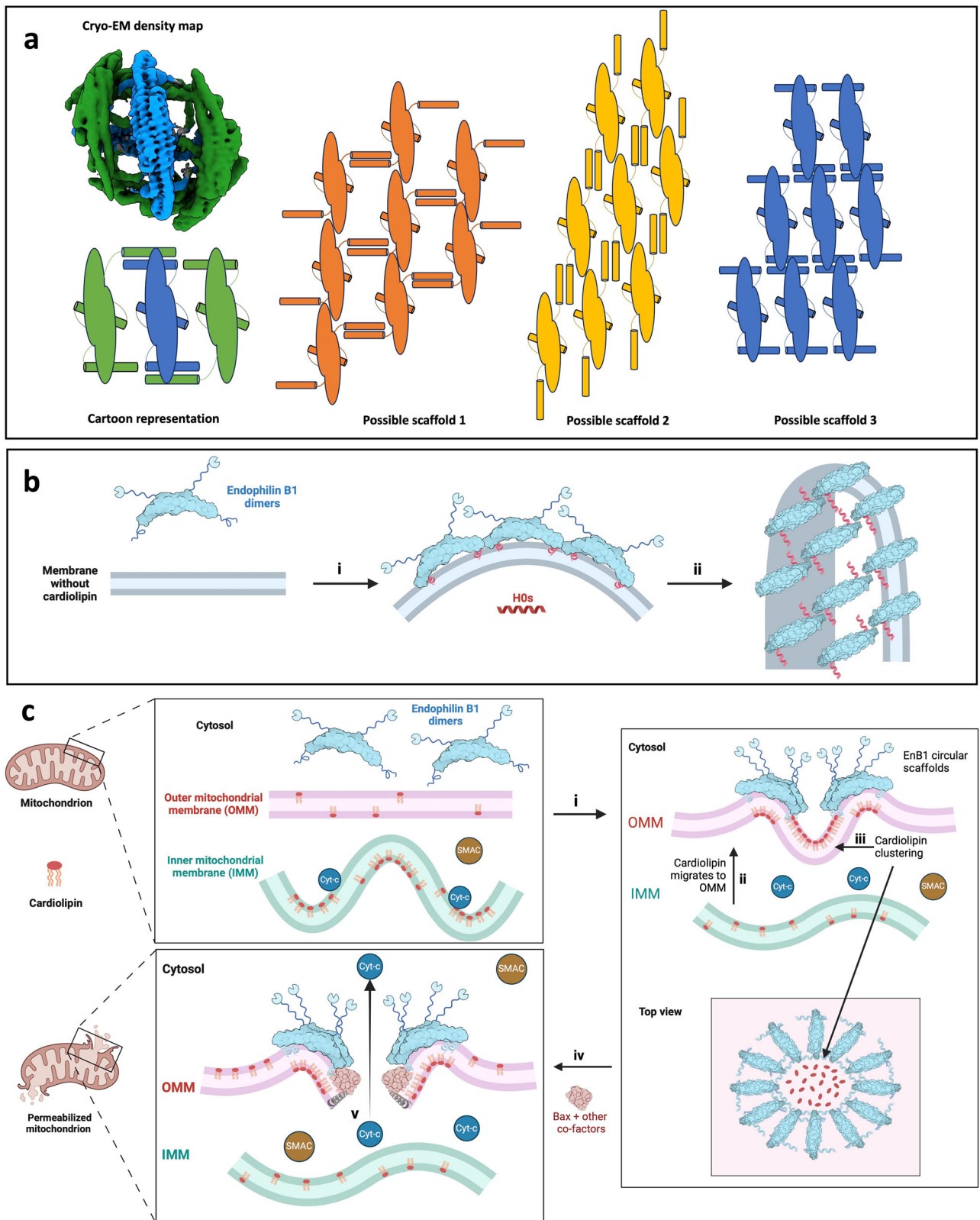

**Fig. 6 | Amphipathic motif assembly promotes endophilin B1 activity at outer-mitochondrial membranes (OMMs) and drives apoptosis. a** Center and side dimers colored in the consensus map and a cartoon representation. Three potential forms of helical scaffolds with different orientations of H0. **b** Upon binding to a membrane that does not contain cardiolipin, endophilin B1 oligomerizes end-to-end (i) to form helical scaffolds that enforce membrane tubulation (ii). **c** Following apoptotic stimuli (i), cardiolipin (a non-bilayer lipid that induces negative curvature) migrates to the OMM (ii). The inner mitochondrial membrane (IMM) is destabilized leading to cytochrome-c (Cyt-c) dissociation. Endophilin B1 binds the OMM and oligomerizes side-to-side into circular scaffolds that cause local negative curvature and further clustering of cardiolipin (iii). Bax and other co-factors bind cardiolipin-rich sites (iv). Bax oligomerization induces leakage of Cyt-c from the intermembrane space (v), which triggers a proteolytic cascade, culminating in cell death. Created in BioRender. Thorlacius, A. (2025) https://BioRender.com/q66x841.

cleavage at 4 °C with gentle agitation. His$_6$-tagged SUMO protease and uncleaved protein were removed by applying the filtered sample to a Nickel affinity column and collecting the unbound fraction. Cleaved protein was concentrated and passed over a Superdex 200 Increase 16/600 GL column (Cytiva; Supplementary Fig. 8a) using size-exclusion chromatography buffer (20 mM HEPES, 150 mM NaCl, 0.5 mM TCEP, pH 8.0). SDS-PAGE analysis was used to estimate the purity of the sample (Supplementary Fig. 8b). Pure endophilin B1 (≥99% w/w) was concentrated to 2 mg/ml, flash frozen in liquid N$_2$ and stored at −70 °C until use.

### Expression and purification of MSP2N2

His$_6$-MSP2N2 (Addgene plasmid #29520)[86] was transformed into BL21(DE3) cells and expressed and purified as previously described with some modifications[87]. Bacteria were grown in terrific broth (TB) and induced with 0.2 mM IPTG at OD$_{600}$ = 0.5, followed by expression at 37 °C for 3 h. Cells were harvested and re-suspended in cell lysis buffer (50 mM Tris, 150 mM NaCl, 20 mM Imidazole, and 1% Triton X-100, pH 8.0). Before lysis, cOmplete™ Protease Inhibitor Cocktail (Sigma-Aldrich) and Dnase were added. Cells were lysed at 35 kPSI using a cell disruptor (Constant Systems), and the lysate centrifuged at 30,600 × g for 60 min. Clarified lysate was applied to the gravity column containing Ni-NTA affinity resin pre-equilibrated with lysis buffer. The column was washed with 10 column volumes each of the following wash buffers: (1) 40 mM Tris–HCl, 300 mM NaCl, 1% Triton X-100, pH 8.0. (2) 40 mM Tris–HCl, 300 mM NaCl, 50 mM sodium cholate, 20 mM Imidazole, pH 8.0. (3) 40 mM Tris–HCl, 300 mM NaCl, 40 mM Imidazole, pH 8.0. His$_6$-MSP2N2 was eluted with elution buffer (40 mM Tris–HCl, 300 mM NaCl, 400 mM Imidazole, pH 8.0). The eluted protein was dialyzed into 20 mM Tris–HCl, 100 mM NaCl, pH 8.0 buffer, concentrated, and finally passed over a HiLoad 16/600 Superdex 200 pg column (Cytiva) using sizing buffer (40 mM Tris–HCl, 100 mM NaCl, pH 8.0) for size-exclusion chromatography (Supplementary Fig. 8c). SDS–PAGE analysis was used to estimate the purity of the sample (Supplementary Fig. 8d). Pure His$_6$-MSP2N2 (≥99% w/w) was concentrated to 1.5 mg/ml, flash frozen in liquid N$_2$ and stored at −70 °C until use.

### Nanodisc preparation

Lipid stocks dissolved in chloroform, purchased from Avanti® Polar Lipids, consisting of 18:1 DOPS (1,2-dioleoyl-sn-glycero-3-phospho-L-serine) and 14:0 cardiolipin (1′,3′-bis[1,2-dimyristoyl-sn-glycero-3-phospho]-glycerol) were mixed and dried under N$_2$ gas to form dry lipid films. These were stored under vacuum overnight to ensure full removal of the solvent. Lipid films were solubilized in standard nanodisc buffer (20 mM Tris–HCl, 100 mM NaCl, 0.5 mM EDTA, pH 7.4) supplemented with 1.5% DDM and sonicated at 37 °C for 45 min. His$_6$-MSP2N2 was incubated with DDM-solubilized lipids at a molar ratio of 1:11:98 (His$_6$-MSP2N2:cardiolipin:DOPS) at 25 °C for 1 h. Samples were dialyzed against standard nanodisc buffer (1:1000) for up to 30 h (dialysis buffer was replaced at least 3 times). The sample was concentrated and run on a Superdex 200 Increase 10/300 GL using standard nanodisc buffer. Fractions containing His$_6$-MSP2N2 were collected and incubated with endophilin-B1 at a molar ratio of 1:20 (His$_6$-MSP2N2:Endophilin B1) for 1 h at 4 °C. The sample was concentrated and run on a Superdex 200 Increase 10/300 GL using sizing buffer (Supplementary Fig. 9a and b). Fractions containing both His$_6$-MSP2N2 and endophilin-B1 were identified by western blot analysis using anti-endophilin B1 polyclonal (Goat; Invitrogen; Supplementary Fig. 9c) and anti-polyHis monoclonal primary antibodies (Mouse; Sigma Aldrich; Supplementary Fig. 9d) followed by alkaline phosphatase-conjugated anti-goat or anti-mouse secondary antibodies (Rockland). Membranes were stained using 1-Step™ NBT/BCIP Substrate Solution (Thermo Scientific).

### Cryo-EM sample preparation and data collection

SEC fractions containing endophilin-B1 and MSP2N2 were pooled and concentrated. Three microliters were applied to glow-discharged holey carbon grids (Quantifoil Micro Tools GmbH). Grids were blotted at 4 °C and 95% humidity using filter paper (Whatman®) and plunge-frozen in liquid ethane using a Vitrobot Mark IV (Thermo Fisher Scientific). Vitrified grids were screened at the Cryo-EM Uppsala facility using 200 kV Glacios (Thermo Fisher Scientific) equipped with a Falcon 3EC direct electron detector (Thermo Fisher Scientific). Cryo-EM data were collected to confirm sample quality and produce initial 3D reconstructions. Endophilin B1-decorated nanodiscs data was collected at SciLifeLab using a Titan Krios G3i (Thermo Fisher Scientific) operated at 300 kV, equipped with a K3 Bio-Quantum direct electron detector (Gatan Inc.) and energy filter using 20 eV slit, at ×130,000 nominal magnification (Supplementary Table 1). Soluble endophilin B1 data was collected at the 3D-EM facility at Karolinska Institutet using a Titan Krios G3i operated at 300 kV, equipped with a cold-FEG (Thermo Fisher Scientific), a K3 BioQuantum direct electron detector (Gatan Inc.) and energy filter using 10 eV slit, at ×165,000 nominal magnification.

### Cryo-EM data processing

Movies from Glacios data collections were imported into cryoSPARC followed by patch motion correction, patch CTF correction, and discarding of low-quality micrographs[60]. Manually picked particles were filtered using 2D classification, and 2D classes were used for template picking. Junk particles were removed through multiple rounds of 2D classification followed by ab-initio reconstruction and non-uniform refinement[88].

Movies from Krios data collections were imported, patch motion corrected and patch CTF corrected in cryoSPARC. Initial picking was performed using template picking, and 2D classes were generated using the final electron density map from the small-scale Glacios data collection on the same sample. Picked particles were filtered using the NCC score and then extracted using a box size of 512$^2$ pixels. These particles were Fourier-cropped to 128$^2$ pixels for initial 2D classification and ab-initio reconstruction. Following the 2D classification and 3D reconstruction, the particles were re-extracted at 512$^2$ pixels and Fourier-cropped to 256$^2$ pixels for further classification using several rounds of heterogeneous ab-initio and heterogeneous refinements. Final classes were refined using non-uniform refinement. Masks for individual endophilin B1 dimers were generated in UCSF ChimeraX and used in local refinements in cryoSPARC (Supplementary Table 1)[89].

### CryoDRGN

Particle sets were taken from different parts of the data processing "timeline" and analyzed with cryoDRGN[61,62]. Before training, each particle set was Fourier-cropped to 128$^2$ pixels in cryoDRGN. Larger and more heterogeneous data was processed over 30–60 epochs using the heterogeneous ab-initio reconstruction function with default architecture (3 × 256), and smaller, less heterogeneous data was processed using the train_vae function, initially with the default architecture and later with a larger architecture (3 × 1024). Particle sets were split into groups based on k-means filtering using the jupyter-notebook script provided in the cryoDRGN software package. Groups were separately re-imported to cryoSPARC for validation using ab-initio reconstruction and non-uniform refinement. Movies were created using a variation of the workflow described in[90] (https://github.com/Guillawme/).

### Model building

The AlphaFold model for endophilin B1 was processed in PHENIX[52,53,91,92]. The model was rigid-body docked into the density maps in UCSF ChimeraX and refined in Coot and Servalcat[89,93–95]. Model validation was performed using MolProbity (Table 1)[96].

### Small-angle X-ray scattering measurements

All SAXS measurements were carried out on the BM29 beamline (ESRF, Grenoble, France). All measurements were performed with 100% beam intensity at a wavelength of 0.9918 Å (12.5 keV). Initial data processing was

**Article**

performed automatically using the EDNA pipeline. See Table 2 for other details of SAXS measurements.

## Small-angle scattering data processing

SAXS profiles ($I(q)$) were processed using the ATSAS and BioXTAS RAW software suites[97,98]. The influence of structural factors on scattering curves was negligible due to low protein concentration. To calculate molar absorption coefficient ($\varepsilon$), molecular mass, and scattering length density (SLD) from sequences, ProtParam, Peptide Property Calculator, and SLD calculator were used (Table 2). Distance distribution functions $P(r)$ and regularized $I(q)$ were obtained using GNOM, which utilizes Indirect-Fourier Transform (IFT)[99]. Values for $R_G$ and $I(0)$ (Table 2) were calculated from $P(r)$, and using Guinier approximations. Model electron density fit and ab-initio electron density maps were created using DENSS[100].

## Liposome preparation

Lipid stocks dissolved in chloroform, 18:1 DOPS (1,2-dioleoyl-sn-glycero-3-phospho-L-serine), 18:1 DOPE (1,2-dioleoyl-sn-glycero-3-phosphoethanolamine), 18:1 cardiolipin (1',3'-bis[1,2-dioleoyl-sn-glycero-3-phospho]-glycerol) and cholesterol, were mixed and dried under $N_2$ gas to form dry lipid films. These were stored under vacuum overnight to ensure full removal of the solvent. Lipid films were solubilized in a 100 mM calcein (Merck) solution and extruded through a 1 µm filter. Calcein-encapsulated liposomes were separated from free calcein by gel filtration using G-50 Sephadex® resin (Cytiva) and sizing buffer. The encapsulation efficiency for each batch was determined by comparing liposomes alone (negative control) with liposomes in the presence of 1% Triton X-100 (Alfa Aesar), which fully permeabilizes liposomes. The fluorescence signal from positive controls was at least 10-fold higher than that of negative controls in all subsequent assays.

## Calcein release assay

Calcein-encapsulated liposomes with different lipid compositions were added to endophilin B1 (final concentration 500 nM or 5 µM) in a 96-well plate and an increase in calcein fluorescence was monitored in a CLARIOstar Plus plate reader (BMG Labtech) at excitation wavelength 482 nm and bandwidth 16 nm, emission wavelength 530 nm and bandwidth 40 nm with the dichroic mirror set to 504 nm. Sample fluorescence was determined every 5 min over a 60-min period with shaking for 10 s prior to measuring. Negative controls consisted of calcein-encapsulated liposomes without protein. Positive controls consisted of calcein-encapsulated liposomes and 1% Triton X-100, which represented the maximum release of calcein. Negative and positive controls were included in each run and results were calculated using the following equation: % Calcein release = $(F_{exp} - F_{neg})/(F_{pos} - F_{neg})$, where $F_{exp}$ is the fluorescence of the sample, $F_0$ the average fluorescence of the negative controls and $F_{pos}$ the average signal from positive controls.

## Statistics and reproducibility

Data presented in Fig. 5 represents sample distribution and mean from 18 (a) or ≥6 experiments (b), respectively. The statistically significant difference between different experimental conditions was determined using a Student's $t$-tests. $P$-values were calculated with a confidence interval of ≥ 99%.

## Reporting summary

Further information on research design is available in the Nature Portfolio Reporting Summary linked to this article.

## Data availability

Raw data for Fig. 5 can be downloaded from Figshare (https://doi.org/10.6084/m9.figshare.28239134). Motion-corrected micrographs are available for download from EMPIAR with the public accession code: EMPIAR-12470[101]. Additional data and material supporting the findings of this manuscript are available from the corresponding author upon reasonable request.

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

## Acknowledgements

Data was collected at the Cryo-EM Swedish National Facility funded by the Knut and Alice Wallenberg, Family Erling Persson and Kempe Foundations, SciLifeLab, Stockholm University, at the Karolinska Institutet 3D-EM facility (https://ki.se/cmb/3d-em) and at Cryo-EM Uppsala. Cryo-EM reconstruction was enabled by the Davinci cluster at Uppsala University ICM with generous technical assistance from Prof. Filipe Maia and Dr. Daniel Larsson. CryoDRGN computation was enabled by the Berzelius resource provided by the Knut and Alice Wallenberg Foundation at the National Supercomputer Centre. We want to thank Prof. Stefan Knight for valuable feedback during the model building. We also acknowledge the European Synchrotron Radiation Facility (ESRF) for the provision of synchrotron radiation facilities and we would like to thank Dr. Petra Pernot for assistance and support in using beamline BM29. We would like to thank the Swedish Research Council (2021-05-423 ASL) for funding this project.

## Author contributions

A.S.-L. conceptualized the work. A.S.-L. and A.T. designed biochemical and Cryo-EM experiments, analyzed the results and prepared the manuscript. A.T. performed all biochemical assays, cryo-EM experiments, data processing and model building. A.T. and O.S. performed membrane

permeabilization assays. A.T. and M.R. designed and performed SAXS experiments. M.R. analyzed the SAXS results.

## Funding

## Competing interests
The authors declare no competing interests.
