## [Transparent Peer Review file · Communications Biology]

Peripheral membrane protein endophilin B1 probes, perturbs and permeabilizes lipid bilayers

Corresponding Author: Dr Anna Sundborger-Lunna

Version 0:

Reviewer comments:

Reviewer #1

(Remarks to the Author)

Thorlacius et al. present the structure of the BAR-protein endophilin B1 bound to lipids that are likely assemble into a bicelle. The lipid-protein particles contain up to six endophilin B1 dimers. Single particle analysis reveals that endophilin B1 can adopt two different conformations depending on the position on the ellipsoid bicelle. Interestingly, amphipathic helices that interact with the membrane are resolved. Both endophilin B1 conformations differ in their bending and the arrangement of the amphipathic helices, suggesting the formation of a variety of possible lattices. Finally, biochemical assays show that endophilin B1 can permeabilize membranes, which may be linked to the role in apoptosis.

This study is of good technical quality and provides new insight into the function of a BAR protein. The newly described ability of an endophilin to form lipoprotein particles is intriguing. The following points should be addressed:

1. One conclusion from the permeabilization experiments is that cardiolipin is required for this activity (Fig 5). However, there is little difference compared to a lipid mix with only cholesterol. As a control, 100% DOPC is used, a 55% DOPS /45% DOPE mix would be a better reference. Increasing CL from 5% to 20% does not have an effect. It is also possible that DOPS contributes to endophilin B1 liposome binding and permeabilization.

The exact lipid composition that was used for nanodisc preparation should be given (l. 441-443)

2. It may be advisable to combine Results and Discussion. In particular, the finding that endophilin forms a lipoprotein particle and displaces MSP2N2 should be mentioned earlier. It is actually quite confusing that the authors refer to nanodisc-bound endophilin (l. 147, l. 191) throughout most of the manuscript although the data in Fig 1-3 and E1 is incompatible with the presence of the original nanodisc in the particles.

“BAR protein bound to a nanodisc (l15)” should also be removed from the abstract.

3. It would be interesting to compare the possible scaffolds (Fig 6a) with the assemblies observed in helical reconstructions.

4. In Fig 4, the number of endophilins in the different classes are difficult to identify in the cryoDRGN and cryoSPARC reconstructions. E.g. in Fig 4b, it looks more like four “bars” in cryoDRGN and three in cryoSPARC? A model of docked endophilins B1 for each class would be helpful.

5. In Table 1, the parts of the N-terminal sequence that actually are helical should be marked. As the amphipathic helix apparently is shorter than the 31 residues shown (l. 158), also the charge the helical segment rather than the entire N-terminal stretch would be relevant (l 45-47).

6. What is the evidence for a circular EnB1 assembly (Fig 6c, right panel)? Could membrane permeabilization occur by the formation of lipoprotein particles similar to those investigated in this study?

Reviewer #2

(Remarks to the Author)

In this manuscript, Thorlacius and colleagues present cryo-EM analysis of Endophilin B1, a BAR-domain containing protein involved in autophagy and control of mitochondrial morphology. The authors have previously characterized the ability of Endophilin B1 to tubulate membranes containing Cardiolipin, a lipid found in mitochondria. This previous work was a more “traditional” approach that characterized the structures of helical filaments on membrane tubules. This work extends that study to the use of lipid nanodiscs as a membrane scaffold, allowing for higher-resolution analysis and delineation of structural features that were not feasible using other approaches. The authors also provide biochemical evidence to support the hypothesis that Endophilin B1 can rupture lipid bilayers, which is modulated by the lipid composition of the membrane.

Overall, this manuscript is technically sound and provides important structural insights into the “active” states of Endophilin B1 on membranes. The use of lipid nanodiscs is technically innovative and has provided unexpected findings. I very much appreciate that the authors released their maps and models, which made visualization of their main takeaways much easier throughout this review. I had some doubts about the model building from looking at the figures, but after looking at the maps, the orientation of all of the bilayer-inserting regions of the protein are more readily apparent. In general, I am highly enthusiastic about this manuscript.

I have three major comments and several minor comments that I would like to see addressed.

Major comments.

1. EMPIAR deposition

I would strongly advocate for the authors to deposit their dose-weighted, aligned micrographs into EMPIAR. There are very few datasets like this that have been generated, and even fewer that have been released publicly. I think this would be a great benefit to the community as a shared resource.

2. Further structural refinement

This appears to be a challenging dataset, and the authors should be commended for pushing the resolution of their middle dimer and side dimer structures past 4 Ang resolution. I have two outstanding questions that should be addressed.

I. Dimer-Dimer interactions. Considering that dimer-dimer interactions mediated by the H0 domain is a major component of this work, the fact that multiple focused refinements centered on the various dimer-dimer interfaces was not performed is puzzling. Have the authors tried to improve the resolution of important interfaces? For example, Figure 2c and 2d explicitly show a dimer-dimer interface. Relatedly, perhaps some of the membrane-embedded helices could be better resolved if the focus masks didn't contain the entire dimer. I understand that, especially in cryoSPARC, refinements don't do well with very small masks. However, perhaps masks that have a similar amount of protein density, but are centered on the important interactions within the membrane, could improve the resolution of some of these regions.

II. There appears to be no attempt at using the overall symmetry of the particle to increase resolution. Is there symmetry in the particle? If so, have you tried symmetry expansion, followed by focused refinement?

III. FSC Curves. The FSC curves in Extended Data 3 show a deep dip in the region in the 8-6 Ang range, which is indicative of a strong residual beam tilt. Did the authors perform NU Refinement with higher order aberration correction? If so, what is the estimated beam tilt? If not, please re-run and ensure that this is not what is causing the aberrant FSC curve shape.

3. Displacement of MSP2N2. The authors say that MSP2N2 is present in the nanodiscs they purified as judged by western blot, but then say that they believe it has been displaced by Endophilin. The only evidence to support this is that they do not see ordered density for MSP2N2 in the cryo-EM map. Without biochemical support, another answer is that the amorphous belt is simply averaged out of their maps. Having worked extensively with nanodiscs, I will say that they often do strange things and that I wouldn't be surprised by either scenario.

However, looking at the maps, I do understand the conclusion that the authors have come to. The only observable density seems to be Endophilin B1 helical domains on the sides, and a thin, amorphous density that likely corresponds to the lipid head groups on the top and bottom.

Reading the paper, however, I don't believe that this conclusion was presented well by the authors. I believe that they should make some sort of additional figure that shows the “belt” of Endophilin B1 helical domains around the nanodisc. A comparison to what a traditional nanodisc would look like would also be helpful.

Minor comments.

1. The lipid mixture used in the cryo-EM sample prep for the nanodisc should be listed in the main text, especially since lipid composition is the crux of the biochemical data presented in the latter half of the manuscript.

2. The coloring in Figure 2 is problematic. In Figures 2a and 2b, the Center and Side dimer are colored blue and green, respectively. However, in figures 2c and 2d, the authors decide to highlight the interaction of a blue Center Dimer with a red side dimer. It would have been easier to follow if the blue/green coloring for those particular dimers was followed through all of figure 2.

3. On line 112 and 133 the authors say “Both elution peaks correspond to particles 113 significantly larger than “naked” nanodiscs.” Looking at Extended Data Figure 1a, this doesn't appear to be the case for the 1:2 ratio, which seems to overlap primarily with the nanodisc peak.

-Rick Baker

UNC Chapel Hill

Version 1:

Reviewer comments:

Reviewer #1

(Remarks to the Author)

The authors have carefully addressed my comments. Congratulations on this very interesting paper, which should be published without further delay.

Reviewer #2

(Remarks to the Author)

I have no additional comments for this manuscript. It is acceptable for publication.

Rebuttal 1

Response to reviewer's comments

Reviewer #1 (Remarks to the Author):

Thorlacius et al. present the structure of the BAR-protein endophilin B1 bound to lipids that are likely assemble into a bicelle. The lipid-protein particles contain up to six endophilin B1 dimers. Single particle analysis reveals that endophilin B1 can adopt two different conformations depending on the position on the ellipsoid bicelle. Interestingly, amphipathic helices that interact with the membrane are resolved. Both endophilin B1 conformations differ in their bending and the arrangement of the amphipathic helices, suggesting the formation of a variety of possible lattices. Finally, biochemical assays show that endophilin B1 can permeabilize membranes, which may be linked to the role in apoptosis. This study is of good technical quality and provides new insight into the function of a BAR protein. The newly described ability of an endophilin to form lipoprotein particles is intriguing.

The following points should be addressed:

1. One conclusion from the permeabilization experiments is that cardiolipin is required for this activity (Fig 5). However, there is little difference compared to a lipid mix with only cholesterol. As a control, 100% DOPC is used, a 55% DOPS/45% DOPE mix would be a better reference. Increasing CL from 5% to 20% does not have an effect. It is also possible that DOPS contributes to endophilin B1 liposome binding and permeabilization.

We completely agree with the reviewer and have rewritten the manuscript to reflect that the major factor to influence endophilin B1 permeabilization behavior is membrane charge (which is due to the presence of DOPS, definitely), but that packing defects known to be mediated by non-bilayer forming lipids, like cardiolipin, may contribute also.

LINES 267-271

2. The exact lipid composition that was used for nanodisc preparation should be given (l. 441-443)

We completely agree and this is now provided.

LINES 107, 467-468

3. It may be advisable to combine Results and Discussion.

We refer this question to the editor.

4. In particular, the finding that endophilin forms a lipoprotein particle and displaces MSP2N2 should be mentioned earlier. It is actually quite confusing that the authors refer

to nanodisc-bound endophilin (l. 147, l. 191) throughout most of the manuscript although the data in Fig 1-3 and E1 is incompatible with the presence of the original nanodisc in the particles.

We agree with the reviewer's comments and have changed the manuscript to reflect this. We now refer to "endophilin B1 decorated nanodiscs" as "lipoprotein complexes" or "endophilin B1 decorated bicelles". Undecorated nanodiscs are still referred to as "nanodiscs". We decided to first describe the process of obtaining the density map and building the atomic model before describing the structure as an "endophilin B1 decorated bicelle" as this observation was made possible only after model building.

LINE 15: Changed "nanodisc" to "lipid bicelle".

LINES 79, 116, 119, 120, 121, 142: Changed "nanodisc" to "lipoprotein complexes"

LINE 125, 126, 128, 142: Changed "endophilin B1 decorated nanodisc" to "endophilin B1 decorated bicelles"

LINES 140-142: Added "This, together with the smaller observed diameter compared to MSP2N2 nanodiscs, is evidence that the particles in the cryo-EM reconstruction are not endophilin decorated nanodiscs but rather lipoprotein complexes consisting of endophilin B1 decorated bicelles."

LINES 160, 197, 207, 209, 212, 214, 216, 305, 322, 323, 330: Changed "nanodisc" to "bicelle".

5. It would be interesting to compare the possible scaffolds (Fig 6a) with the assemblies observed in helical reconstructions.

We completely agree with the reviewer's comment. The way dimers assemble into helical scaffolds is heavily influenced by the membrane lipid composition. We show that in the differences in tube diameter and the distinct number of lipid-protein interfaces. We now show that on local patches of membrane, amphipathic helices (AHs) have distinct modes of insertion (fig. 6). We believe that these diverse modes are presented in the helical reconstruction but that the low resolution prevents us from capturing these organizations. Docking distinct AH organizations into the helical scaffold map was attempted, but due to the low resolution, it was not possible to discern specific modes. We are planning to study this further and address this in a future publication.

6. In Fig 4, the number of endophilins in the different classes are difficult to identify in the cryoDRGN and cryoSPARC reconstructions. E.g. in Fig 4b, it looks more like four "bars" in cryoDRGN and three in cryoSPARC? A model of docked endophilins B1 for each class would be helpful.

We want to thank the reviewer for this excellent suggestion and have modified figure 4 to better illustrate this.

Please find updated figure 4 below!

Figure modifications: Side panel (b) has been added and lower panels (c-k) have been modified to include atomic models overlaid the density maps.

- In Table 1, the parts of the N-terminal sequence that actually are helical should be marked. As the amphipathic helix apparently is shorter than the 31 residues shown (l. 158), also the charge the helical segment rather than the entire N-terminal stretch would be relevant (l. 45-47).

This is another great suggestion from the reviewer.

Please find the updated table 1 below!

H0	Sequence	
Endophilin A1	-----MSVAGLKKQFHKATQKVSEKVG	22
Endophilin A2	-----MSVAGLKKQFYKASQLVSEKVG	22
Endophilin A3	-----MSVAGLKKQFHKASQLFSEKIS	22
Endophilin B1	MNIMDFNVKK LAADAGTFLSRVQFTEEKLG	31
Endophilin B2	---MDFNMKKLASDAGIFFTRAVQFTEEKFG	28

Figure modifications: the helical region of endophilin B1 is now marked in red, while the tip of H0 is bold.

8. What is the evidence for a circular EnB1 assembly (Fig 6c, right panel)? Could membrane permeabilization occur by the formation of lipoprotein particles similar to those investigated in this study?

The circular assembly is speculation based on the notion that cardiolipin promotes negative curvature and that clustering of cardiolipin therefore may cause membrane "pits" to form and that pit edges may attract endophilin B1 due to its concave shape. This our model that we will test in future studies.

Reviewer #2 (Remarks to the Author):

In this manuscript, Thorlacius and colleagues present cryo-EM analysis of Endophilin B1, a BAR-domain containing protein involved in autophagy and control of mitochondrial morphology. The authors have previously characterized the ability of Endophilin B1 to tubulate membranes containing Cardiolipin, a lipid found in mitochondria. This previous work was a more "traditional" approach that characterized the structures of helical filaments on membrane tubules. This work extends that study to the use of lipid nanodiscs as a membrane scaffold, allowing for higher-resolution analysis and delineation of structural features that were not feasible using other approaches. The authors also provide biochemical evidence to support the hypothesis that Endophilin B1 can rupture lipid bilayers, which is modulated by the lipid composition of the membrane.

Overall, this manuscript is technically sound and provides important structural insights into the "active" states of Endophilin B1 on membranes. The use of lipid nanodiscs is technically innovative and has provided unexpected findings. I very much appreciate that the authors released their maps and models, which made visualization of their main takeaways much easier throughout this review. I had some doubts about the model building from looking at the figures, but after looking at the maps, the orientation of all of the bilayer-inserting regions of the protein are more readily apparent. In general, I am highly enthusiastic about this manuscript.

I have three major comments and several minor comments that I would like to see addressed.

Major comments.

1. EMPIAR deposition. I would strongly advocate for the authors to deposit their dose-weighted, aligned micrographs into EMPIAR. There are very few datasets like this that have been generated, and even fewer that have been released publicly. I think this would be a great benefit to the community as a shared resource.

We agree and have submitted our motion corrected micrographs to EMPIAR and added information about this to our Data availability statement.

LINES 433-434

2. Further structural refinement. This appears to be a challenging dataset, and the authors should be commended for pushing the resolution of their middle dimer and side dimer structures past 4 Ang resolution. I have two outstanding questions that should be addressed.

I. Dimer-Dimer interactions. Considering that dimer-dimer interactions mediated by the H0 domain is a major component of this work, the fact that multiple focused refinements centered on the various dimer-dimer interfaces was not performed is puzzling. Have the authors tried to improve the resolution of important interfaces?

We want to thank the reviewer for this great comment. We tried multiple times, but the membrane-bound helices appear "mobile". Generating masks for single H0s and even the H0-lipid interface resulted in overfitting. We tried using two H0s from the same dimer and anti-parallel H0s from different dimers. Generating masks that encompassed multiple H0s at the same time was also a mess. We tried using cryoDRGN, cryoSPARC 3D flex, as well as cryoSPARC 3DVA in combination with masking, but those also did not work well. We have added a comment about focused refinement to the text.

LINES 169-170

For example, Figure 2c and 2d explicitly show a dimer-dimer interface. Relatedly, perhaps some of the membrane-embedded helices could be better resolved if the focus masks didn't contain the entire dimer. I understand that, especially in cryoSPARC, refinements don't do well with very small masks. However, perhaps masks that have a similar amount of protein density, but are centered on the important interactions within the membrane, could improve the resolution of some of these regions.

Exactly. H0s are too small to mask alone. They are also flexible and moving around. We tried one more time to mask a dimer-dimer interface, but see no improved resolution of the region. cryoDRGN predictions support the conclusion that they are simply too small and mobile to effectively mask.

II. There appears to be no attempt at using the overall symmetry of the particle to increase resolution. Is there symmetry in the particle? If so, have you tried symmetry expansion, followed by focused refinement?

Symmetry expansion followed by focused refinement did not markedly improve the global resolution. We decided to use C1 symmetry to avoid the whole symmetry vs. false symmetry debate.

III. FSC Curves. The FSC curves in Extended Data 3 show a deep dip in the region in the 8-6 Ang range, which is indicative of a strong residual beam tilt. Did the authors perform NU Refinement with higher order aberration correction? If so, what is the estimated beam tilt? If not, please re-run and ensure that this is not what is causing the aberrant FSC curve shape.

Yes, we did. This can be done using the Global CTF refinement job in cryoSPARC. We re-ran the jobs and strong residual beam tilt is not what is causing the aberrant shape of the FSC curve. Others have reported similar dips when working with membrane proteins that are surrounded by a lipid belt or have flexible regions. This has been discussed in the creators of cryoSPARC in their forum several times, e.g.: <https://discuss.cryosparc.com/t/extra-bump-in-fsc/2170/3>

3. The authors say that MSP2N2 is present in the nanodiscs they purified as judged by western blot, but then say that they believe it has been displaced by endophilin. The only evidence to support this is that they do not see ordered density for MSP2N2 in the cryo-EM map. Without biochemical support, another answer is that the amorphous belt is simply averaged out of their maps. Having worked extensively with nanodiscs, I will say that they often do strange things and that I wouldn't be surprised by either scenario. However, looking at the maps, I do understand the conclusion that the authors have come to. The only observable density seems to be Endophilin B1 helical domains on the sides, and a thin, amorphous density that likely corresponds to the lipid head groups on the top and bottom. Reading the paper, however, I don't believe that this conclusion was presented well by the authors. I believe that they should make some sort of additional figure that shows the "belt" of Endophilin B1 helical domains around the nanodisc. A comparison to what a traditional nanodisc would look like would also be helpful.

We did indeed struggle with how to best describe our findings. Our final endophilin B1-containing structure does not contain any density that could correspond to MSP2N2, suggesting that endophilin B1 “kicks off” the scaffolding protein. However, the structure only consists of a small subset of all the particles present in the sample, which means that there are still nanodiscs present in the sample (that can be detected by WB). We have generated a new figure (Extended data figure 10) to better illustrate our results and rephrased the text in the discussion.

LINES 311-317

Please find updated extended data figure 10 below!

Extended data figure 10. Top row: The structure of endophilin B1 decorated lipoprotein particles (PDB ID: 9G2R) docked into our consensus map of endophilin B1 decorated lipoprotein particles (EMD-50981) with the amphipathic motifs highlighted in red. Middle row: The NMR structure of MSP1D1 (shown in blue) nanodiscs (PDB ID: 6CLZ) docked into the consensus map of endophilin B1 decorated lipoprotein particles (EMD-50981). The bilayer in the 6CLZ structure has a similar diameter and width of the lipid bilayer as that displayed in our consensus map. Bottom row: The superimposed structures of endophilin B1 decorated lipoprotein particles (PDB ID: 9G2R) and NMR structure of MSP1D1 (PDB ID: 6CLZ) docked into the consensus map of endophilin b1 decorated lipoprotein particles (EMD-50981). Superimposing the two structures shows that the amphipathic motifs of endophilin B1 (red) in the lipoprotein complex occupy the same positions as MSP1D1 (blue) does in the nanodiscs structure.

Minor comments.

1. The lipid mixture used in the cryo-EM sample prep for the nanodisc should be listed in the main text, especially since lipid composition is the crux of the biochemical data presented in the latter half of the manuscript.

This is now stated in the text.

LINE 107

2. The coloring in Figure 2 is problematic. In Figures 2a and 2b, the Center and Side dimer are colored blue and green, respectively. However, in figures 2c and 2d, the authors decide to highlight the interaction of a blue Center Dimer with a red side dimer. It would have been easier to follow if the blue/green coloring for those particular dimers was followed through all of figure 2.

We have updated figure 2 according to the excellent feedback.

Please find updated figure 2 below!

Figure modifications: the colors of amphipathic helices were changed.

3. On line 112 and 133 the authors say "Both elution peaks correspond to particles significantly larger than "naked" nanodiscs." Looking at Extended Data Figure 1a, this doesn't appear to be the case for the 1:2 ratio, which seems to overlap primarily with the nanodisc peak.

We completely agree with the reviewer and have changed the text accordingly.

LINES 112-114: *changed to "The sample containing more endophilin B1 (1:10) corresponds to a larger particle size than the sample with less endophilin B1 (1:2; Extended Data Fig. 1a)."*